# Specific features and assembly of the plant mitochondrial complex I revealed by cryo-EM

Heddy Soufari[1], Camila Parrot[1], Lauriane Kuhn [2], Florent Waltz [1✉] & Yaser Hashem [1✉]

Mitochondria are the powerhouses of eukaryotic cells and the site of essential metabolic reactions. Complex I or NADH:ubiquinone oxidoreductase is the main entry site for electrons into the mitochondrial respiratory chain and constitutes the largest of the respiratory complexes. Its structure and composition vary across eukaryote species. However, high resolution structures are available only for one group of eukaryotes, opisthokonts. In plants, only biochemical studies were carried out, already hinting at the peculiar composition of complex I in the green lineage. Here, we report several cryo-electron microscopy structures of the plant mitochondrial complex I. We describe the structure and composition of the plant respiratory complex I, including the ancestral mitochondrial domain composed of the carbonic anhydrase. We show that the carbonic anhydrase is a heterotrimeric complex with only one conserved active site. This domain is crucial for the overall stability of complex I as well as a peculiar lipid complex composed of cardiolipin and phosphatidylinositols. Moreover, we also describe the structure of one of the plant-specific complex I assembly intermediates, lacking the whole $P_D$ module, in presence of the maturation factor GLDH. GLDH prevents the binding of the plant specific P1 protein, responsible for the linkage of the $P_P$ to the $P_D$ module.

---

[1] Institut Européen de Chimie et Biologie, U1212 Inserm, Université de Bordeaux, 2 rue R. Escarpit, F-33600 Pessac, France. [2] Plateforme protéomique Strasbourg Esplanade FRC1589 du CNRS, Université de Strasbourg, Strasbourg, France. ✉email: florent.waltz@etu.unistra.fr; yaser.hashem@inserm.fr

Complex I is the largest multimeric enzyme of the respiratory chain, composed of more than 40 protein subunits[1]. Fourteen are strictly conserved proteins subunits, vestige of complex I of bacterial origin[2]. The additional subunits, referred to as supernumerary subunits, were acquired during eukaryotes evolution[3]. Part of these additional subunits are conserved among other eukaryotes and play essential roles for the structure, function and the association with the other respiratory chain complexes, e.g., to assemble into respirasome in animals[4]. Recently, several high resolution 3D structures of the complete mitochondrial complex I of opisthokonts were determined by cryo-EM in mammalian species[5–7] and in the aerobic yeast *Yarrowia lipolytica*[8,9], revealing the organization of their additional specie-specific subunits. However, in plants, even though extensive biochemical characterization was conducted[10,11], high-resolution structures of mitochondrial complex I are yet to be derived. Early negative staining studies[11] revealed the presence of a large additional membrane attached domain, absent from animal and yeast species, hinting at the peculiar structure and composition of the plant complex I.

In order to obtain a high-resolution structure of the plant mitochondrial complex I and its additional subunits, we purified mitochondria from *Brassica oleracea var. botrytis*, a close relative to the model plant Arabidopsis (both belong to the group of Brassicaceae plants), as previously described[12] (see "Methods" section). We recorded cryo-EM images of membrane complexes purified from sucrose gradient (see "Methods" section), corresponding to mitochondrial complex I. After particle sorting (see "Methods" section) we obtained cryo-EM reconstructions of two main complex I states: the full plant mitochondrial complex I, as well as reconstruction of a complex I assembly intermediate, without the $P_D$ module, and with the plant-specific assembly factor GLDH[13] (Supplementary Fig. 1). Focused refinement and particle polishing were performed in RELION3[14] (see "Methods" section), allowing reconstruction of the full complex I at 3.7 Å resolution, focused classification on the membrane arm and $P_P$ module yielding 3.4 Å. The $P_D$ module being more flexible produced more scant densities reporting lower resolutions (Supplementary Figs. 1, 2). The cryo-EM maps allowed the determination of many well-resolved features and clear side-chain densities (Supplementary Fig. 3) that enabled modeling of the 45 proteins of the plant mitochondrial complex I (Supplementary Table 1), as well as several characteristic ligands that were directly identified from the density: the eight canonical FeS clusters of the matrix arm, the FMN of the 51 kDa subunit, the NADPH molecule in the 39 kDa subunit, three cardiolipins two phosphatidylinositol and one phosphatidylethanolamine (Supplementary Fig. 3). Ubiquinone was also clearly visible in the Q module (Supplementary Fig. 4).

## Results and discussion

**General description**. The plant mitochondrial complex I has the classical open L-shape formed by the matrix and membrane arms (Fig. 1). In the matrix arm, electrons are transferred from NADH to ubiquinone along the FeS clusters with distances similar to what has been described in bacteria and opisthokonts mitochondrial complex I (Supplementary Fig. 5). The membrane part, composed of the proximal ($P_P$) and distal ($P_D$) modules relative to the matrix arm, where proton pumping takes place, is also conserved. Those highly conserved features are part of the 14 minimal protein subunits conserved in bacteria and other mitochondrial complex I[1]. The additional mitochondria specific proteins enhance the volume and mass of the complex I compared to bacteria. In plant, 25 proteins shared with aerobic yeast or mammals are present. Similarly to the previously obtained complex I structure[5,7,9,15], the supernumerary subunits, specific

to mitochondrial complex I, form a shell around the core subunits, adding nearly 400 kDa of proteins to the conserved subunits (Supplementary Fig. 5). These supernumerary subunits are mainly composed of α-helices and are largely interconnected with the core subunits. In the Q module of the matrix arm, our reconstructions allowed us to clearly visualize ubiquinone in the pocket formed by Nad1, PSST and Nad7 (Supplementary Fig. 4). Our results confirm the recently observed ubiquinone position described for *Y. lipolytica*[8]. During 3D classification of the mature complex, only one major conformation was observed. This would indicate that the plant complex I would be similar to the yeast complex I, where similar conformations are observed in both active and deactive states[9], as opposed to mammalian complexes, where movement of the matrix arm relative to the membrane arm is observed between the states[5].

**Carbonic anhydrase: an ancestral component of mitochondrial complex I**. The main feature of the plant complex I is the presence of an additional globular matrix-exposed domain, bound to the membrane arm. It holds an overlapping position with the NDUFA10/42-kDa subunit found in mammalian complex I[5]. This domain is formed by a trimeric complex, the carbonic anhydrase (CA). CA are zinc-containing enzymes that catalyze the reversible hydration of $CO_2$ to $HCO_3^-$, whose roles are usually carbon fixation and pH maintenance[16]. The carbonic anhydrase activity evolved independently several times during evolution, resulting in several CA enzyme families. The plant mitochondrial CA is part of the gamma-carbonic anhydrase (γCA) family, which was originally identified in the archaeon *Methanosarcina thermophila*[17,18]. Here, the overall structure of the γCA is conserved: it is a trimer composed of characteristic left-handed beta-helix monomers, forming triangular prisms (Fig. 2a). However, in contrast to the archaeal enzyme, the γCA present in plant complex I is an heterotrimer[19]. Indeed, in Arabidopsis five γCA genes were identified. Three have highly conserved catalytic domains and are termed γCA1-3, similar to the archaeal enzyme, while the two other show less homology to their prokaryotic homologs especially in the catalytic domain and were thus termed γCA-like proteins[20,21]. Our reconstruction shows that the γCA domain is formed by one copy of γCAL, built as γCAL1 representing the most proximal subunit to the membrane part and two copies of γCA. For the two γCA subunits, one was clearly identified as γCA1 but the scant density on the second, which could be explained by compositional heterogeneity, did not allow to clearly distinguish between γCA1 and γCA2, and was therefore built as γCA1 (Fig. 2a and Supplementary Fig. 6). Each subunits have long N and C-terminal extensions interconnecting the 3 subunits and contacting the membrane subunits of complex I (Fig. 2c). These extensions also serve as anchors for the γCA domain by interacting with the plant specific P2 protein N-terminal part which is anchored in the membrane. The C-terminal part of the canonical Nad6 protein also contacts the CA domain stabilizing it. γCA is an enzyme, where one histidine of one subunit and two histidine of the adjacent subunit contribute to coordinate a zinc atom. However, as γCAL proteins lack two of those three histidines, it impairs the formation of the zinc coordination with both γCA subunits. Hence the only conserved active site is at the interface of the two γCAs, which we confirmed by visualizing both the zinc and an additional density most likely corresponding to $HCO_3^-$ (Fig. 2b and Supplementary Fig. 6). This strongly suggests that the γCA domain associated with complex I is indeed active, even if a role in $HCO_3^-$ fixation and/or transport cannot be excluded, similarly to what is observed in cyanobacterial CcmM[22]. However, the function of complex I-associated γCA is not well understood. It was suggested to play a

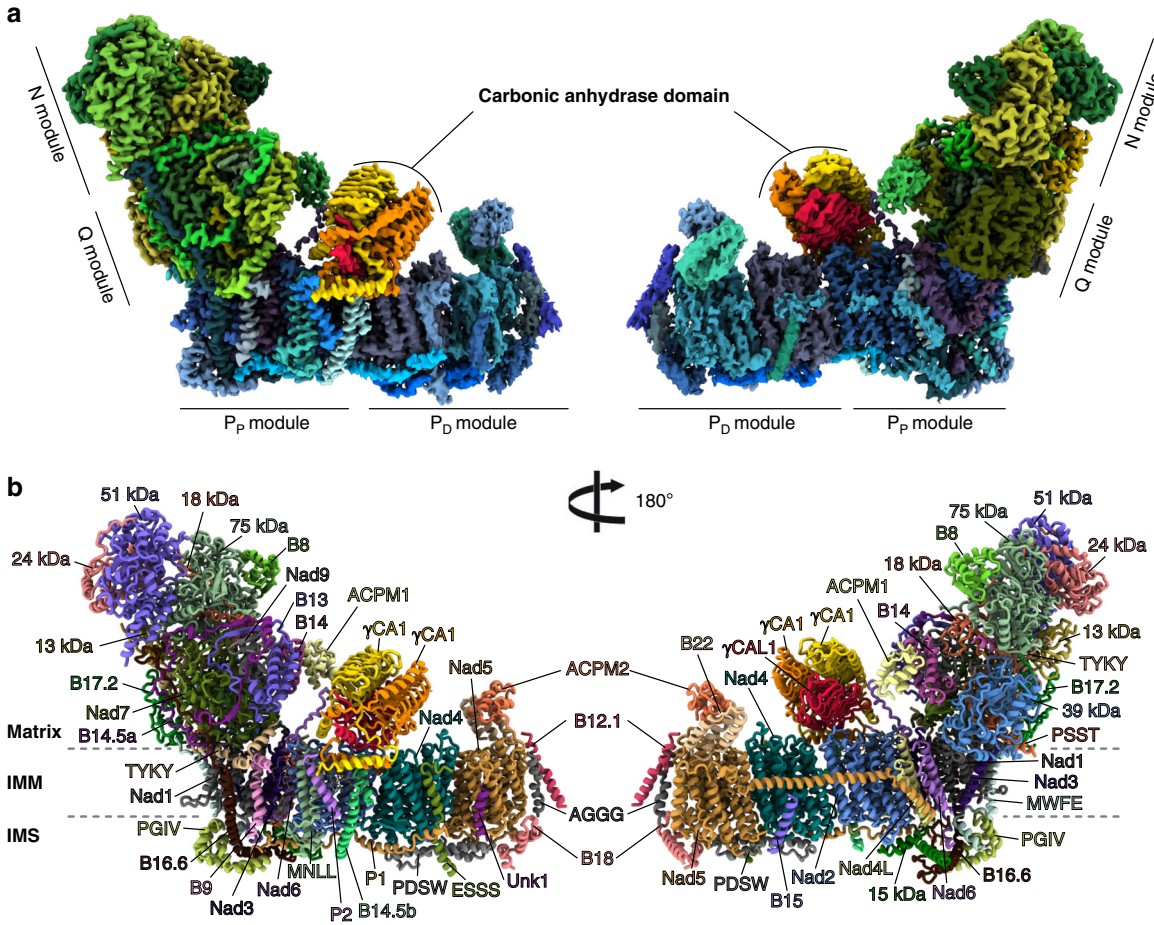

**Fig. 1 Overall structure of the plant mitochondrial complex I. a** Composite cryo-EM map of the complete plant complex I. The membrane part is displayed in blue shades, the matrix arm is displayed in green shades and the γ-carbonic anhydrase module is displayed in orange shades. Individual N, Q, $P_D$, and $P_P$ modules are indicated. **b** The resulting atomic model, each individual 45 proteins are annotated and displayed with a different color. IMM stands for inner mitochondrial membrane, IMS inter-membrane space.

role in complex I assembly as γCA is found in early assembly intermediates of the complex and is essential for complex I formation[23]. Indeed, embryo development of γCA mutants is strongly delayed and seed development arrested before maturation. Moreover complemented mutants with inactive γCA variants is sufficient to complement the mutant phenotype, showing that carbonic anhydrase activity is not essential[20]. Recently it was shown that carbonic anhydrase activity is also present in photosynthetic cyanobacterial complex I, but is carried out by a different type of CA enzyme, positioned at the end of the $P_D$ module that would contribute to proton generation[24]. However, in plant mitochondrial complex I, the only conserved active site is positioned at the most distal position relative to the membrane. Nevertheless, one cannot exclude its contribution to proton generation, although its effect would be relatively modest.

Interestingly, the CA domain is in tight interaction with a peculiar lipid complex positioned below the latter and between the plant specific P2 and proteins B14.5b and Nad2, close to the Nad2–Nad4 junction (Fig. 2d). This lipid complex is composed of one cardiolipin coordinated by two phosphatidylinositol. Each inositol heads sandwich the glycerol linker of the cardiolipin and stabilize it. This lipid block was also observed in the assembly intermediate (see below) and appears to be very stable. Given the essential role of lipids for complex I activity, this lipid block might be crucial for the plant complex I efficiency and stability, which might be why CA mutants are so heavily affected[8,25]. Altogether, our analysis strongly suggest that the γCA is essential

for the membrane part formation and stability, acting as an essential architectural factor that would remain even in mature complex.

**Assembly intermediate of the plant complex I.** During 3D classification, two classes of complex I lacking the most distal proton pumping components of the membrane domain—Nad4 and Nad5 and their associated additional protein subunits—were found to naturally accumulate in our sample (Supplementary Fig. 1). One of the two classes presented an additional globular density on the intermembrane space exposed side of complex I. Such density wasn't observed in full complex I classes. This class was thus attributed to an assembly intermediate of complex I (Fig. 3). In plants, such complex was already described to accumulate naturally. Indeed, extensive biochemical analyses identified complex I* as an assembly intermediate lacking the most distal membrane components and presenting an additional factor, GLDH (L-Galactono-1,4-lactone dehydrogenase). GLDH was identified in our mass spectrometry data (Supplementary Table 1), along with the other components of complex I. Likewise γCA, GLDH is an enzyme catalysing the last enzymatic step of the ascorbate biosynthetic pathway in plants[26]. It was never found in the full complex I, however, it was shown to associate with complex I intermediates, complex I* being the largest, as well as with smaller complex intermediates that were not observed here[23]. Moreover, in *gldh* knock-out mutant, complex I

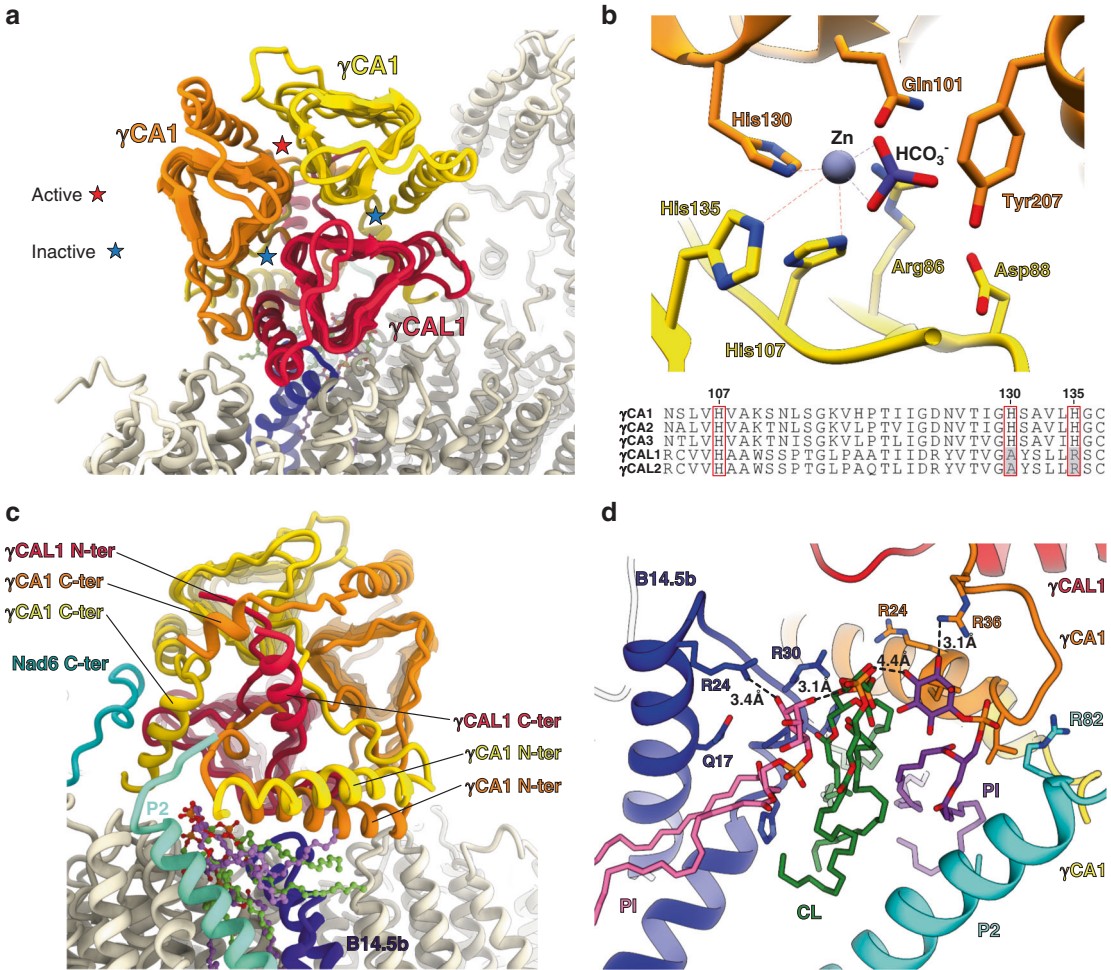

**Fig. 2 Hetero-trimeric γ-carbonic anhydrase of the plant mitochondrial complex I. a–c** Front and back view of the carbonic anhydrase showing individual subunits and contacting proteins. Active and inactive sites are shown in **a**. **b** Close up of the γCA1/γCA1 only conserved active site, showing the coordinate zinc by the three histidine residues between the two γCA1 subunits. Sequence alignment of γCA and γCAL highlight the catalytic histidines. Surrounding residues, Tyr207, Gln101, Arg86, are also shown and might be involved in $HCO_3^-$ coordination. **c** Back view of the carbonic anhydrase, highlighting the N and C-ter extensions of the γCA and γCAL, as well as the proteins allowing to anchor the whole domain to the membrane arm. **d** The lipid block composed of one central cardiolipin and two phosphatidylinositol molecules is shown in a close-up view, with surrounding residues. The lipids are sandwiched between the carbonic anhydrase domain, the plant-specific protein P2, Nad2, and B14.5. Both inositol head groups contribute to coordinate the cardiolipin. CL stands for cardiolipin and PI for phosphatidylinositol.

is undetectable, which hinted its role as a plant specific assembly factor[13,27]. Using focused classification and refinement, we were able to derive an intermediate-resolution reconstruction on the additional density corresponding to GLDH which was sufficient to attribute the density to the protein. Importantly, the complex I binding domain of GLDH is resolved at high resolution (3.8), thus validating our assignment (Fig. 3b–d). In this assembly intermediate, GLDH interacts with B14.5b—that is slightly displaced compared to the mature complex—and the 15 kDa subunit. Moreover the whole carbonic anhydrase is slightly shifted toward the missing $P_D$ module (Fig. 3c). We also successfully identified the N-terminal tail of GLDH bound to the maturing complex I, between B14.5b and 15 kDa preventing the plant specific protein P1 from docking to the $P_P$ module. P1 appears to functionally replace the NDUFB5 subunit, which is absent in plants[23] (Supplementary Fig. 5). In the GLDH context the P1 path is blocked in the assembly intermediate due to B14.5b movement and N-ter tail of GLDH (Fig. 3). Thus, it appears that GLDH prevents the binding of P1 to the $P_P$ module, until its release, which will then allow association with the $P_D$ module. The clear mechanism of action of GLDH in the context of the

assembly intermediate is however unclear. It was previously shown that GLDH can perform its original enzymatic activity, however this activity is not required from complex I formation, its functions are thus not linked[13]. Hence, in plants, GLDH might act as a stabilization factor of complex I assembly intermediates that would be released just before the final step of $P_D$ module docking to complex I*. It is unlikely that P1 alone could remove GLDH, thus this process would probably involve unknown additional trans-acting factor(s). Similarly, in yeast the assembly factor NDUFAF2 is released from the assembly intermediate by the joint action of three accessory subunits[8].

In conclusion, our high-resolution cryo-EM structures of the plant mitochondrial complex I, reveals plant-specific and ancestral features of this respiratory complex (Fig. 4). Indeed, γCA was identified as a complex I component in all Viridiplantae, including algae of the Chlorophyceae class[11,28–31], as well as in *Euglena gracilis*, a photosynthetic protozoan related to trypanosomes[32] and possibly *Acanthamoeba castellanii*, a protozoan ameboid[33] and in non-photosynthetic organisms (e.g., Tetrahymena and Reclinomonas)[33] (Fig. 4). Given the wide presence of the γCA in eukaryotes—members of the γCA are found in all

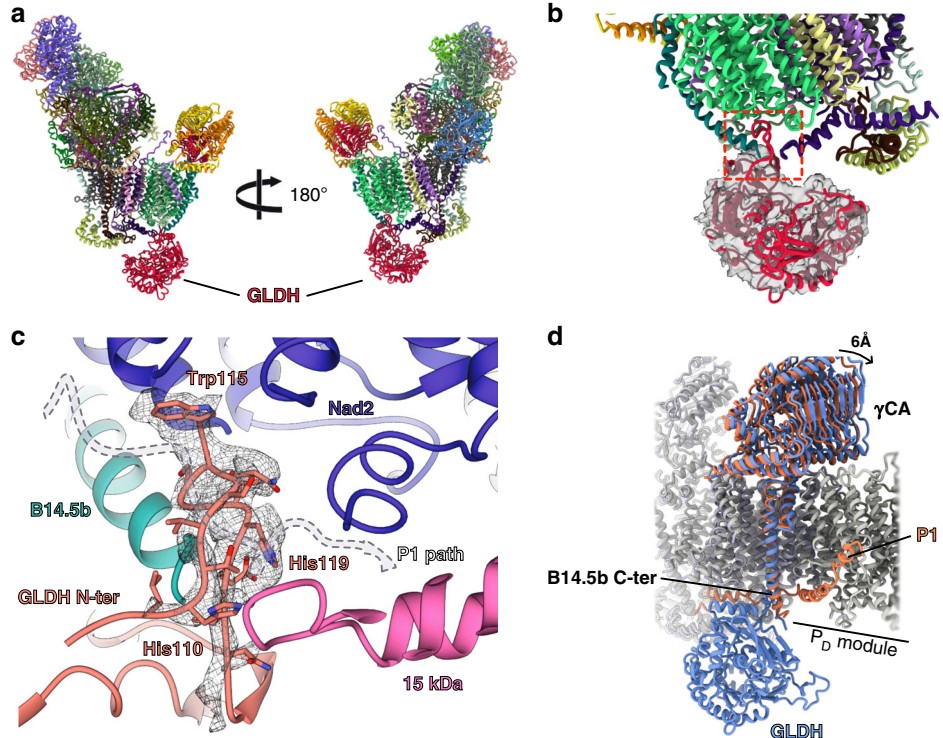

**Fig. 3 GLDH is a plant-specific assembly factor of mitochondrial complex I. a** Atomic model of the assembly intermediate, the whole $P_D$ module is missing and GLDH contacts the $P_P$ module on the intermembrane side. **b** GLDH is shown in cut-out and filtered density. The interaction point is highlighted and a zoomed view is shown in **c**. **c** Focused view of the interaction point of GLDH with the complex. The N-terminal part of GLDH was identified thanks to specific aromatic residues and is shown in its density. The N-ter part of GLDH, as well as the deviation of B14.5b block P1 path which is shown in dashed lines. **d** Overall movement in the assembly intermediate compared to the mature complex, B14.5b C-terminal part is deviated by 4 Å and the whole γCA domain is shifted by 6 Å toward the missing $P_D$ module, GLDH and absence of the P1 protein are also highlighted. Coral correspond to mature complex I and blue to assembly intermediate.

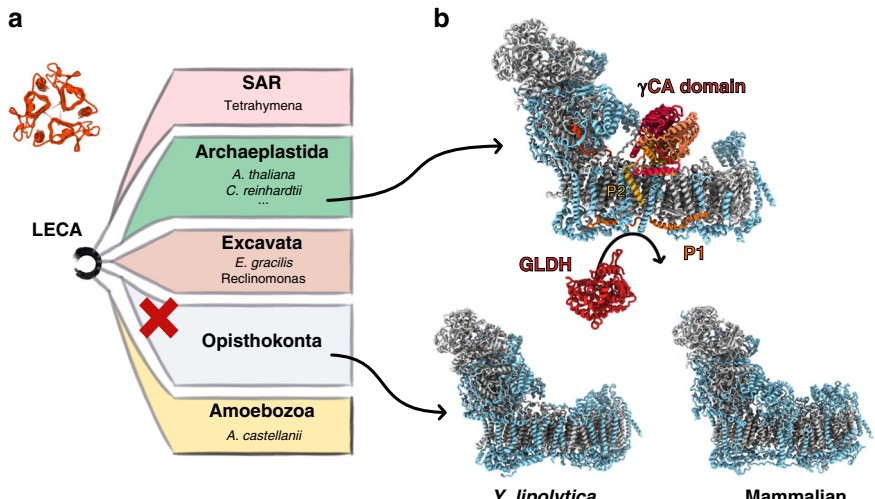

**Fig. 4 Plant mitochondrial complex I specific subunits. a** Schematic representation of the five major eukaryote lineages. Organisms where γ-carbonic anhydrase is proposed to be part of mitochondrial complex I are listed next to their respective lineage. The red cross indicate the loss of mitochondrial γ-carbonic anhydrase in opisthokonta. **b** Comparison of mitochondrial complex I structures. Core subunits are shown in light gray and supernumary subunits in light blue. Plant specific proteins are highlighted in red-orange colors. *Y. lipolytica* (PDB:6RFR) and mammalian (PDB:6G2J) complex I do not have the γ-carbonic anhydrase domain.

eukaryotic lineages except opisthokonts—we present for the first time the structure of a mitochondrial complex I featuring this ancestral component. Moreover, we describe a yet unobserved plant-specific assembly intermediate, shedding light into the respiratory complex I maturation.

## Methods

**Mitochondrial complex I purification**. Cauliflower (*Brassica oleracea* var. botrytis) mitochondria were purified as previously described[12]. Quickly, fresh cauliflower inflorescence tissue was blended in extraction buffer containing 0.3 M mannitol, 30 mM sodium pyrophosphate (10.$H_2O$), 0.5% BSA, 0.8% (w/v) polyvinylpyrrolidone-25, 2 mM beta-mercaptoethanol, 1 mM EDTA, 20 mM ascorbate and 5 mM

cysteine, pH 7.5. Lysate was filtered and clarified by centrifugation at $1500 \times g$, 10 min at 4 °C. Supernatant was kept and centrifuged at $18,000 \times g$, 15 min at 4 °C. Organelle pellet was re-suspended in wash buffer (0.3 M mannitol and 10 mM phosphate buffer, 1 mM EDTA, pH 7.5) and the precedent centrifugations were repeated once. The resulting organelle pellet was re-suspended in wash buffer and loaded on a single-step 30% Percoll gradient (in wash buffer without EDTA) and run for 1h30 at $40,000 \times g$. Mitochondria were retrieved, washed two times and flash frozen in liquid nitrogen.

For complex I purification, mitochondria were re-suspended in Lysis buffer (20 mM HEPES-KOH, pH 7.6, 100 mM KCl, 1 mM DTT, 2% n-β-DDM, 1 mM EDTA, supplemented with proteases inhibitors (Complete)) incubated for 15 min in 4 °C. Lysate was clarified by centrifugation at $30,000 \times g$, 20 min at 4 °C. The supernatant was loaded on a 10–50% sucrose gradient buffer (20 mM HEPES-KOH, pH 7.6, 50 mM KCl, 1 mM DTT, 0.2% n-β-DDM, 1 mM EDTA, supplemented with proteases inhibitors (Complete)) and run for 16 h at $75,000 \times g$. The fraction corresponding to complex I was collected, pelleted and re-suspended in final resuspension buffer (same as sucrose gradient buffer without sucrose and only 0.1% n-β-DDM).

**Grid preparation.** Four microliter of the samples at a concentration of 1 μg/μl was applied onto Quantifoil R2/2 300-mesh holey carbon grid, coated with thin home-made continuous carbon film and glow-discharged (3 mA for 20 s). The sample was incubated on the grid for 30 s and then blotted with filter paper for 2.5 s in a temperature and humidity controlled Vitrobot Mark IV (T = 4 °C, humidity 100%, blot force 5) followed by vitrification in liquid ethane.

**Single particle cryo-electron microscopy data collection.** Data collection was performed on a Talos Arctica instrument (Thermofisher Company) at 200 kV using the SerialEM software for automated data acquisition. Data were collected at a nominal underfocus of −0.5 to −2.5 μm at a magnification of ×36,000 yielding a pixel size of 1.11 Å. Micrographs were recorded as movie stack on a K2 direct electron detector (GATAN Company), each movie stack were fractionated into 65 frames for a total exposure of 6.5 s corresponding to an electron dose of 45 ē/Å2.

**Electron microscopy image processing.** Drift and gain correction and dose weighting were performed using MotionCorr2[34]. A dose weighted average image of the whole stack was used to determine the contrast transfer function with the software Gctf[35]. The following process has been achieved using RELION 3.0[14]. Particles were picked using the general model of crYOLO 1.5[36]. Output box files from crYOLO were imported in RELION and particles were extracted. After reference-free 2D classification, 650,844 particles were extracted with a box size of 360 pixels and binned threefold, resulting in a 120 pixels box size and used for 3D classification into 6 classes (Supplementary Fig. 1). CryoSPARC[37] generated ab-initio cryo-EM map, low-pass filtered to 30 Å, was used as an initial reference for 3D classification. One subclass depicting high-resolution features have been selected for the mature complex I refinement with 65,018 particles. After Bayesian polishing a focused refinement has been performed using mask excluding the $P_D$ module yielding, respectively, 3.7 and 3.47 Å resolution. A second subclass containing the maturation factor GLDH composed of 155,644 particles was selected for further focused 3D classification, using a loose soft mask on the GLDH area. The 36,513 particles of GLDH enriched subclass were extracted for refinement and reached 3.8 Å. Determination of the local resolution of the final density map was performed using ResMap[38].

**Structure building and model refinement.** The atomic model of the plant mitochondrial complex I was built into the high-resolution maps using Coot, Phenix, and Chimera. Atomic models from *Y.lipolytica* (6RFR) and *M.musculus* (6G2J) were used as starting points for protein identification and modelisation. Similarly to Waltz & Soufari 2019[12], as the genome of cauliflower is not sequenced, and the closest fully sequenced member of the family (*Brassica oleracea* subsp. oleracea) is poorly annotated, to facilitate comprehension and analysis we positioned Arabidopsis proteins in the cauliflower map. Still, protein sequence identities between members of the Brassicaceae family are higher than 90%, thus facilitating proteomics identification of cauliflower proteins. The online SWISS-MODEL service was used to generate initial models for bacterial and mitochondria conserved r-proteins. Models were then rigid body fitted to the density in Chimera[39] and all subsequent modeling was done in Coot[40]. Extensions were built has polyalanine and mutated to the adequate sequences. A combination of regularization and real-space refine was performed in Coot for each proteins. The global atomic model was then subjected to real space refinement cycles using *phenix.real_space_refine* PHENIX[41] function, during which protein secondary structure, Ramachandran and side chain rotamer restraints. Several rounds of refinement (manual in Coot and automated using the *phenix.real_space_refine*) were performed to obtain the final models, which were validated using the built-in validation tool of PHENIX, based on MolProbity. For the assembly intermediate, the model was built with the mature complex as template. Refinement and validation statistics were summarized in Supplementary Table 2.

**Proteomic and statistical analyses of mitochondrial complex I composition.** Mass spectrometry analyses of the complex I fractions were performed at the Strasbourg-Esplanade proteomic platform and performed as previously[42]. In brief, proteins were trypsin digested, mass spectrometry analyses and quantitative proteomics were carried out by nanoLC-ESI-MS/MS analysis on a QExactive+ (Thermo) mass spectrometer. Data were searched against the TAIR *A. thaliana* database with a target-decoy strategy (release TAIRv10, 27281 forward protein sequences), and home-made *Brassica oleracea* database extracted from UniProtKB (Swissprot+TrEMBL) including Brassica sub-taxonomy (release 2017_10, 59050 forward protein sequences). Proteins were validated respecting FDR < 1% (False Discovery Rate) and quantitative label-free analysis was performed through in-house bioinformatics pipelines.

**Figure preparation.** Figures featuring cryo-EM densities, as well atomic models were visualized with UCSF ChimeraX[43] and Chimera[39].

**Reporting summary.** Further information on research design is available in the Nature Research Reporting Summary linked to this article.

## Data availability

The cryo-EM maps of plant mitochondrial complex I have been deposited at the Electron Microscopy Data Bank (EMDB): EMD-11614 for the mature complex, EMD-11513 for the focused reconstruction, EMD-11615 for the assembly intermediate. The corresponding atomic models been deposited in the Protein Data Bank (PDB) under the accession 7A23 for the mature complex and 7A24 for the assembly intermediate. Mass spectrometric data have been deposited to the ProteomeXchange Consortium via the PRIDE partner repository with the dataset identifier PXD017847.

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

## Acknowledgements

This work has benefitted from the facilities and expertize of the Biophysical and Structural Chemistry platform (BPCS) at IECB, CNRS UMS3033, Inserm US001, University of Bordeaux. We thank A. Bezault for assistance with the Talos Arctica electron microscope. We thank, J. Chicher and P. Hamman of the Strasbourg Espanade proteomic analysis for the proteomic analysis. This work was supported by a European Research Council Starting Grant (TransTryp ID:759120) and a Agence Nationale de la Recherche (ANR) grant [MITRA, ANR-16-CE11-0024-02] to Y.H.

## Author contributions

F.W., H.S., and Y.H. designed and coordinated the experiments. F.W. purified the mitochondria and mitochondrial complex I. H.S. and C.P. prepared grids. H.S. acquired the cryo-EM data and processed the cryo-EM results with F.W. H.S. and F.W. built the atomic models. L.K. performed the mass-spectrometry experiments. F.W., H.S., and Y.H. interpreted the structure. F.W., H.S., C.P., and Y.H. wrote and edited the manuscript.

## Competing interests

The authors declare no competing interests.
