## [Peer Review File · Nature Communications]

REVIEWER COMMENTS

Reviewer #1 (Remarks to the Author):

The first enzyme in the electron transport chain in Mitochondria is the multi-subunit complex of NADH:ubiquinone oxidoreductase. The oxidation of NADH and transfer of electrons to ubiquinone coupled to proton pumping is an essential step in generation of ATP. The prokaryotic enzymes harbor the core subunits that comprise all the necessary cofactors for the function but eukaryotes have evolved to have additional subunits (also called supernumerary) that directly don't take part in enzymatic reaction but are essential for stability and assembly and some of these subunits have other additional functions. The composition of some of these supernumerary subunits differ between species and with recent advances in cryoEM such differences can be now be visualized. In the manuscript, Soufari et al present structures of complex I from plant mitochondria highlighting unique (or special) features of this enzyme and also present the structure of one of the assembly intermediate.

As these enzymes have a number of subunits and except for 7 of them, the rest are imported into the mitochondria (or nuclear encoded). Thus, an important scientific question is to understand how this complex machinery is assembled using proteins both from mitochondria and nucleus. By presenting the full enzyme and one of the intermediate the manuscript strives to achieve this goal. Though concise with very nice illustrations, I have several queries that when addressed will the make the manuscript much more readable to general audience.

a) In the abstract (which might be too long – not sure if Nature Communications accepts more than 150 words), the authors mention multiple structures. However, in the table describing cryoEM data and extended data 2, only the full complex and the focused refinement of this complex is described (this accounts to one model and two maps). The description of the intermediate is provided and the density showing the GLDH molecule is presented in Figure 3 but no information on how this was docked and refined is presented in methods or table. This will be essential to include as this is one of the salient feature of the manuscript and the map/model of this needs to be deposited.

b) The class/map shown in green (extended data 1) has not been analyzed further but a brief mention of shifted Pd module is written in figure legend. It will be useful to expand on this – for instance, what has happened to transverse helix, is the complex is falling apart, is the matrix arm the same as in other classes etc.,

c) The final step of purification has 0.1% DDM, does this affect the stability of complex and has other detergents been tried to asses the quality of enzyme.

d) In the method section – EM processing needs to be written a bit more clearly. For instance, the 3D classification step (line 357-358) says that "4-fold for 3D into 6 classes for each subunit" is very vague. What was the reference used for this 3D classification. The gray maps do not have high-resolution features and it is understandable that this has been discarded (ED figure 1). The map in yellow, which looks similar to green/blue has been discarded, why was it discarded? Also, when describing focused 3D classification of the assembly intermediate, will be useful to know how it was done (meaning what sort of mask).

e) The clash score of the model is high (30). I tried looking into the validation report from PDB and some of them highlighted are the SF4/FES. Please check if the coordination and geometry of these with their residues is refined properly (chemically correct and not distorted). In this regard, it will be useful to explain bit more on how the modelling and refinement was performed. It is mentioned that phenix software was used in both real and reciprocal space for refinement but how was this done (iteratively or all in reciprocal space and the final in real space?)

Minor points:

- 1) Cryo EM table – symmetry as C1 and not none. The ligand B-factor seems low when compared to protein. Please check
- 2) Figure 1 – instead of IM/OM, it should be matrix and IMS
- 3) Figure 2 – in panel D, density for lipids are not required (shown in ED figure 3). Instead the

interaction between head group of PI and CL can be shown

4) Figure 3 – panel C & D can be swapped so that they follow an order

5) ED figure 1 – A micrograph of the enzyme will be useful. The reason I am interested is because a thin layer of carbon was used to make grids and relatively high concentration (1 ug/ul) was used (for a carbon coated grid). It will be useful to see the distribution. Panels (a,b) are missing in the figure.

6) Please mention the glow discharge time for the grids in the method.

7) ED Figure 2 – Along with FSC 0.143 (for the half-maps), a map vs model FSC is also required. The resolution of this comparison (map vs model) should also be plotted in the graph.

8) ED figure 3 – panel a, the cartoon can be made thin so that the chain trace can be visualized easily. Along with the α -helices, please also include a region with β -strands (example 75 kDa or PSST or 51 kDa subunit). For the same figure, I have a suggestion. Using panels k/l (where the position of ligands are marked), the density of the ligands can be shown in the highlighted region instead of individually as panels e-j.

9) ED figure 4 – will be useful to show the interaction of UBQ with residues (or distances). I don't know the number in plant mitochondria but its equivalent Y144 in yeast (PSST) and N2 cluster in Nad7 can be shown and the distance between the UBQ and N2 mentioned. This will give an indication of how far the UBQ is with respect to the last FES cluster.

10) Please swap ED figure 5 & 6. In the text, ED figure 6 is mentioned first and only then ED figure 5 comes.

11) ED figure 5 – panel a density is not very clear. Please try to reduce the contour to see if the depiction gets better. Also, useful to show the density of equivalent regions at all 3 sites (γ CA/ γ CAL)

12) In methods, line 329, should be n- β -DDM

13) In methods, line 349, "65 frames with a total exposure of 1 s"? Should this be 10 seconds. If the total dose was 45 e/ Å^2 and this was fractionated to 65 frames, then the exposure time should be higher.

14) In methods – mass spectrometry, line 379 - ? ribosome fractions

15) Several typos – ED table 1, 51kDa – FNM instead of FMN; P1/11 kDa – functionally and not functionally; ED figure 4 legend, first line the protein name is PSST and not PPST; in methods line 357 – bow instead of box

Reviewer #2 (Remarks to the Author):

The manuscript entitled "Specific features and assembly of the plant mitochondrial complex I revealed by cryo-EM" by Soufari et al. represents an important and robust advance within the subject of Biochemistry, particularly mitochondria, respiration and metabolism. The data nicely confirm a large number of biochemical studies and add some novel features, particularly on the structure of the special CA domain present in several eukaryotic lineages except Opisthokonta (animals and fungi). The structures also confirm the presence of an assembly intermediate involving the GLDH protein.

The manuscript is clearly written, however, I have some concerns and questions that may help to improve the work:

1-The authors wrote in several parts of the manuscript including the abstract "the plant-specific additional domain composed by carbonic anhydrase proteins". However, the authors also refer that, as it was already published, gamma carbonic anhydrases were found as part of complex I or suggested to be part of complex I in several other eukaryotes. Thus, I think that continue calling them as "plant-specific" is not correct. In fact, this topic was discussed several years ago and the consensus was to name them as "ancestral subunits". I am not quite sure whether this name is the appropriate name because this encloses the idea that in Opisthokonta the domain was lost and we are not absolutely sure whether this domain was lost or never existed in this particular lineage. However, the name of "ancestral subunits" was accepted and certainly it is better than "plant-specific" which we gave at the beginning. Thus, in my opinion, the last sentence of the abstract

and manuscript does not represent really a novelty because it was already proposed.

2-Lane 89 Carbonic anhydrase: specific component of the plant complex I. It is a sub-title; however it is not specific of plants. Thus, I suggest changing the sub title.

3-The authors confirm the proposition of heterotrimeric composition of the CA domain, containing two γ CAs and one γ CAL, proposed in several papers; particularly it is shown that γ CA1 and two γ CAL subunits were found. This particular arrangement of two identical γ CAs and one γ CAL was proposed in Córdoba et al., 2019 (Plant Cell Physiol. 60:986-998). Moreover, the data of this publication suggest that the composition varies throughout development, being the CA2-CA2-CAL heterotrimer the more important in adult plants, mainly leaves and roots (Brassica and Arabidopsis contain exactly the same kind of γ CA subunits with high homology. Something about the conservation of the subunits between both species should be also appropriate as supplementary material, for example). The CA1-CA1-CAL heterotrimer is more important in embryos at the beginning of embryogenesis in Arabidopsis. Since mitochondria were isolated from *B. oleracea* inflorescence which are arrested flower meristematic tissue, did the author obtain data from other tissues? Did the authors find some heterogeneity at this level? Indeed, the identification of these subunits according to the text, figures and supplementary data appears to be very specific and clear. However, in the supplementary data, it is shown γ CA2, γ CA1 and γ CAL1. Why are there two γ CA1 subunits in the model (Figure 2)? Are there models with γ CA2 subunits instead γ CA1 as well? In the extended data Figure 5 d and e, the authors show some indications why the protein detected is γ CA1 and not γ CA3 or γ CA2, however the authors wrote, "Still for the second γ CA (chain q) the scant density did not allow to clearly distinguish between γ CA1 and γ CA2, and was therefore built as γ CA1". The conclusion is not very clear to me although for the chain p was clear that γ CA1 was present. The writing should be changed, in my opinion to avoid confusion.

Brassica has also two γ CAL proteins. For the biochemical data obtained in Arabidopsis, apparently they are always at the same level and in all tissues, without preferences for heterotrimer formation. Then I would expect some heterogeneity although γ CAL1 and γ CAL2 are very similar as the authors mention in the legend of extended data of Figure 5c. Is it possible to guarantee the identity of this subunit (γ CAL1 or γ CAL2?) or are they so similar that they could be confused?

4-Lane 115. The authors concluded that the CA domain of complex I is indeed active as a carbonic anhydrase because they found a density most likely corresponding to $-\text{HCO}_3^-$ bonded together with a Zinc atom to the active site. In my opinion this sentence should be modified. The binding to bicarbonate/ CO_2 was reported previously by Martin et al., 2009, (FEBS Lett. 583:3425-3430). However, the presence of bicarbonate does not mean necessarily that the enzyme is active as carbonic anhydrase but as a bicarbonate binding protein for regulation or to facilitate bicarbonate transport. This activity (i.e. Regulate access of cytosolic bicarbonate in the case of cyanobacteria) was suggested for CcmM protein of the shell of carboxysome in cyanobacteria which share high sequence and structural homology to gamma type carbonic anhydrases and binds bicarbonate but is inactive as a carbonic anhydrase (Cot et al., 2008 J Bacteriol. 190:936-945). Bicarbonate was suggested to enter the carboxysome in a regulated manner and then to be converted to CO_2 by a beta type carbonic anhydrase (CcaA). CcmM also binds RubisCO inside the carboxysome. The structure of the complex formed by CcmM and RubisCO was solved in Wang et al., 2019 (Nature 566:131-135). In 2010, it was discovered that some cyanobacteria species which do not contain a CcaA inside the carboxysome, the CcmM protein was active as carbonic anhydrase (Peña et al., 2010, Proc Natl Acad Sci U S A. 107:2455-2460). Thus, I think that the conclusion is not supported by the data and therefore should be revised.

5-Lane 119. The authors refer to an experiment in which γ ca2 mutants were complemented with inactive variants of γ CA2. They cite a review of Fromm et al. 2016 (Physiol Plant. 157:289-296). but I think this citation should be changed for the original paper of Fromm et al., 2016a (New Phytol. 211:194-207). However, in my opinion this experiment means that carbonic anhydrase activity is not essential but it cannot be concluded that binding to bicarbonate is not essential because in the mutant variants, aminoacids important for bicarbonate binding which are not so clear were not mutated. It would be interesting whether with the presented data the authors can determine what aminoacids are in contact with the bicarbonate molecule. This would add novel important information to the work.

6-The discovery of the lipid complex interacting with the CA domain is very interesting and

represents important data for future work. However, the authors attribute the reason why the CA mutants show strong phenotypes to the interaction with this particular lipid complex (I suppose the strong phenotype referred is lethal phenotype in normal growth conditions showing no complex I formation in ca1ca2 and ca1ca3 double mutants, Fromm et al., 2016a *New Phytol.* 211:194-207, Fromm et al., 2016b, *J Exp Bot.* 67:3079-3093, Córdoba et al., 2016, *J Exp Bot.* 67:1589-1603 and Córdoba et al., 2019, *Plant Cell Physiol.* 6:986-998). This is maybe true, but in my opinion is one of the reasons. The other more important reason according to the proposed assembly mechanism (Ligas et al., 2019 already cited) is that the CA domain is in the first detected intermediate of 200 kDa. Is it possible that in this first 200 kDa intermediate, this particular lipid complex is already attached and allows the binding of Nad2 subunit?

7-It is very interesting the confirmation of the plant-specific assembly intermediate containing the GLDH assembly factor which is not present in the full complex I.

Minor points

Lane 31 "...composed cardiolipin and phosphatidylinositols." It should be "composed of"

Lane 50 The citation is not correct. The text corresponds to the Dudkina et al., which is cited immediately above (citation 11). The citation 12 is Sunderhaus et al., which is the first correlation that the identified gamma CAs formed the extra domain in plants.

Lane 98, the citation 18 is not corresponding with the text. For articles about *Metanosarcina thermophila* Cam the authors could cite Alber et al., 1999, *Biochemistry* 38:13119-13128 and Tripp et al., 2004, *J Biol Chem.* 279:6683-6687. The citation of Parisi et al., 2004, should be cited when the authors mention about the presence of gamma CAs as complex I subunits. For citation 19, I suggest to add Perales et al., 2004, *Plant Mol Biol.* 56:947-957 in which the original data were published. In this paper heterotrimeric CA domain containing a CAL subunit was first proposed.

Lane 121-123. I think the authors should indicate that the carbonic anhydrase associated to photosynthetic complex I is in cyanobacteria which evolved more sophisticated carbon concentration mechanisms than plants.

Lane 123 instead "our complex I" please write mitochondrial complex I

Lane 166, the sentence of the conclusion is misreading.

Lane 172, change "but not in" by "except in"

Eduardo Zabaleta. Instituto de Investigaciones Biológicas, IIB-CONICET-UNMdP, Mar del Plata, Argentina

Reviewer #3 (Remarks to the Author):

Soufari et al report the first structure of a plant complex I structure determined by cryo-electron microscopy single particle analysis. The authors purified the n-DDM solubilized complex from Cauliflower. Cryo-EM single particle analysis revealed two major types of complexes: the complete complex I and an assembly intermediate that misses the PD sub-complex and has the assembly factor GLDH bound to its intermediate space surface. Compared to previously determined complex I structures from yeast and mammals the most striking difference is the carbonic anhydrase complex (CA) bound to the matrix interface of complex I. The assembly factor GLDH is plant specific and provides a first insight into the molecular mechanism of a respiration complex maturation factor.

The EM structure determination appears solid overall, although it is difficult to judge in the complete absence of any raw data and some typical quality measures that are typically provided. The model building into the 3.5-3.8 Å resolution maps was greatly helped by available models of the yeast and mammalian complex-I homologs. It is somewhat unfortunate that the model was built for *Arabidopsis* proteins in the absence of well-annotated cauliflower sequences.

Overall, the most interesting insights of the paper are structural data about the interference of (plant-specific) GLDH with complete complex-I assembly by competition with (plant-specific) P1 binding, and the association of CA to complex-I. Weaknesses of the manuscript are the absence of any biochemical or genetic experiments as well as the model building based on another sequence.

For publication, the authors should either model the cauliflower proteins (preferred) or substantiate the expected sequence variations for the modeled proteins compared to Arabidopsis.

Major points

- It appears odd that the authors modeled complex I subunits based on Arabidopsis sequences, while mass-spectrometry could detect the actual cauliflower proteins. The possibility to perform protein identification by MS strongly suggests that the protein database compiled for MS identification could be used for model building as well.
- Local resolution in a surface representation are standard for large assemblies determined by cryo-EM and not provided here
- Cross resolution of map and atomic map should be provided as a sanity test of atomic model and resolution
- A representative micrograph must be placed in supplement
- Some idea of the unmasked complex (including the micelle) should be provided as supplement
- Cauliflower and Arabidopsis are reported to be 'closely related' – however, no approximation of expected sequence identities is provided.
- The assignment of different paralogs of CA in the EM density appears unreliable: the assignment bases on identification of characteristic side chains – however, the precise sequences of the cauliflower proteins are not even used. Neither local resolution of the EM map nor possible sequence variations suggest that paralogs can be distinguished.
- Extended data figure 3 a-d: without an indication of sequence conservation the in-depth visualization of side chains does not appear warranted despite the clear visibility of large side chains
- Some more detailed discussion of the analogies of animal vs plant subunits would be appreciated. E.g., how do P1 and NDUF5 compare?
- l. 155: what does 'sufficient to attribute the density to the protein' mean. What is the criterion for 'sufficient'? what is near-atomic resolution (atomic would be better than $\sim 2 \text{ \AA}$!).
- l. 157: what does 'slightly displaced' mean?
- L 166/167: sentence is incomplete – and unclear. What are 'most likely ancestral features of this respiratory complex'?
- Short introductory sentences on P_D and P_P module would be appreciated by a broad audience.
- Universal role of carbonic anhydrase is somewhat vaguely suggested in the abstract, but not really explained. Also the term 'ancestral-like' organization is unclear.

minor points

- l. 158: 'and 15 kDa'? something appears to be missing.
- P_D module should be P lowercase D consistently.
- L. 68: it should probably read: "...the NADPH molecule in the 39kDa subunit, ".

A detailed point by point answer to reviewers and a description of changes brought to the manuscript are as follows (comments in blue, our answers in black). In the main text, changes are highlighted in green.

Response to reviewers' comments:

Reviewer #1

Though concise with very nice illustrations, I have several queries that when addressed will make the manuscript much more readable to general audience.

We thank reviewer 1 for his/her positive appreciation of our work.

a) In the abstract (which might be too long – not sure if Nature Communications accepts more than 150 words), the authors mention multiple structures. However, in the table describing cryoEM data and extended data 2, only the full complex and the focused refinement of this complex is described (this accounts to one model and two maps). The description of the intermediate is provided and the density showing the GLDH molecule is presented in Figure 3 but no information on how this was docked and refined is presented in methods or table. This will be essential to include as this is one of the salient feature of the manuscript and the map/model of this needs to be deposited.

The table now includes both models (full and assembly intermediate). A ResMap is also provided in ED Figure 2 for the GLDH reconstruction. The abstract was revised to a shorter size.

b) The class/map shown in green (extended data 1) has not been analyzed further but a brief mention of shifted Pd module is written in figure legend. It will be useful to expand on this – for instance, what has happened to transverse helix, is the complex is falling apart, is the matrix arm the same as in other classes etc.,

It appears that the whole P_D module is flexible, the transverse helix of Nad5 is not visible due to very poor resolution of the whole P_D module. Another explanation would indeed be that we are witnessing complex I is falling apart, which might explain why we had also a lot of broken complexes. A more extended description of these complexes has been added to ED Fig1.

c) The final step of purification has 0.1% DDM, does this affect the stability of complex and has other detergents been tried to assess the quality of enzyme.

In our experiments we only tried two types of detergent, Triton-X100 and DDM. The triton-X100 results are not shown because it failed to purify respiratory complexes altogether. We used DDM because we had previously used it to check plant respiratory complexes integrity through Blue-Native PAGE and knew it would yield intact complex I. Concerning the final step of purification, 0.1% DDM was used to preserve the complexes on the grid. From literature and our experience, higher detergent concentration would hamper the contrast of the cryo-EM micrographs, which could translate in lower resolution of the final reconstructions.

d) In the method section – EM processing needs to be written a bit more clearly. For instance, the 3D classification step (line 357-358) says that “4-fold for 3D into 6 classes for each subunit” is very vague. What was the reference used for this 3D classification. The gray maps do not have high-resolution features and it is understandable that this has been discarded (ED figure 1). The map in yellow, which looks similar to green/blue has been

discarded, why was it discarded? Also, when describing focused 3D classification of the assembly intermediate, will be useful to know how it was done (meaning what sort of mask).

Method section has been revised for more clarity and precision. The yellow class, even though featuring apparent high resolution in ED Figure 1, corresponded to broken complex I (not intermediate neither full) and was thus not used any further in the processing steps. As similarly done for the green class, a more comprehensive description of these reconstructions has been added to ED Fig1.

e) The clash score of the model is high (30). I tried looking into the validation report from PDB and some of them highlighted are the SF4/FES. Please check if the coordination and geometry of these with their residues is refined properly (chemically correct and not distorted). In this regard, it will be useful to explain bit more on how the modelling and refinement was performed. It is mentioned that phenix software was used in both real and reciprocal space for refinement but how was this done (iteratively or all in reciprocal space and the final in real space?)

We warmly thank the reviewer for this comment. It appeared that iron-sulfur cluster were wrongly inserted in the model, which has now been corrected, resulting in an overall improvement of the model statistics. For modelling and refinement, a more comprehensive description of the material and method section has been added to the manuscript.

Minor points:

1) Cryo EM table – symmetry as C1 and not none. The ligand B-factor seems low when compared to protein. Please check

This has been checked and corrected, along with all the other values in the table.

2) Figure 1 – instead of IM/OM, it should be matrix and IMS

This is now corrected.

3) Figure 2 – in panel D, density for lipids are not required (shown in ED figure 3). Instead the interaction between head group of PI and CL can be shown

A zoomed-in view, without the density, is now provided showing interaction between the lipids and the surrounding residues.

4) Figure 3 – panel C & D can be swapped so that they follow an order

The change has been made accordingly.

5) ED figure 1 – A micrograph of the enzyme will be useful. The reason I am interested is because a thin layer of carbon was used to make grids and relatively high concentration (1 ug/ul) was used (for a carbon coated grid). It will be useful to see the distribution. Panels (a,b) are missing in the figure.

ED figure 1 now includes a representative micrograph.

6) Please mention the glow discharge time for the grids in the method.

Glow discharge (3 mA for 20 sec) is now specified in the text.

7) ED Figure 2 – Along with FSC 0.143 (for the half-maps), a map vs model FSC is also

required. The resolution of this comparison (map vs model) should also be plotted in the graph.

Map vs model FSC, extracted from PHENIX validation, are now shown in ED Fig 2.

8) ED figure 3 – panel a, the cartoon can be made thin so that the chain trace can be visualized easily. Along with the α -helices, please also include a region with β -strands (example 75 kDa or PSST or 51 kDa subunit). For the same figure, I have a suggestion. Using panels k/l (where the position of ligands are marked), the density of the ligands can be shown in the highlighted region instead of individually as panels e-j.

We thank the reviewer for this suggestion. The whole figure has been revised to include the reviewer's suggestions. It now includes beta strands and a portion without secondary structure that gives a better view of the resolution.

9) ED figure 4 – will be useful to show the interaction of UBQ with residues (or distances). I don't know the number in plant mitochondria but its equivalent Y144 in yeast (PSST) and N2 cluster in Nad7 can be shown and the distance between the UBQ and N2 mentioned. This will give an indication of how far the UBQ is with respect to the last FES cluster.

ED Figure 4 has been modified to include distances from ubiquinone to N2 cluster and surrounding residues. The distance observed here (26.4A) is nearly identical to what is observed in yeast (26.7A)

10) Please swap ED figure 5 & 6. In the text, ED figure 6 is mentioned first and only then ED figure 5 comes.

Figure 5 and 6 have been swapped.

11) ED figure 5 – panel a density is not very clear. Please try to reduce the contour to see if the depiction gets better. Also, useful to show the density of equivalent regions at all 3 sites (γ CA/ γ CAL)

The density of the original panel has been adjusted, and two panels corresponding to the inactive sites have been added.

13) In methods, line 349, "65 frames with a total exposure of 1 s"? Should this be 10 seconds. If the total dose was 45 e/ \AA^2 and this was fractionated to 65 frames, then the exposure time should be higher.

Exposure time has been corrected, 6.5 sec was the correct time

12) In methods, line 329, should be n- β -DDM

14) In methods – mass spectrometry, line 379 - ? ribosome fractions

15) Several typos – ED table 1, 51kDa – FNM instead of FMN; P1/11 kDa – functionally and not functionally; ED figure 4 legend, first line the protein name is PSST and not PPST; in methods line 357 – bow instead of box

All the minor points have been addressed

Reviewer #2

The manuscript entitled “Specific features and assembly of the plant mitochondrial complex I revealed by cryo-EM” by Soufari et al. represents an important and robust advance within the subject of Biochemistry, particularly mitochondria, respiration and metabolism. The data nicely confirm a large number of biochemical studies and add some novel features, particularly on the structure of the special CA domain present in several eukaryotic lineages except Opisthokonta (animals and fungi). The structures also confirm the presence of an assembly intermediate involving the GLDH protein.

We thank reviewer 2 for this positive appreciation of our work and thorough comments.

The manuscript is clearly written, however, I have some concerns and questions that may help to improve the work:

1-The authors wrote in several parts of the manuscript including the abstract “the plant-specific additional domain composed by carbonic anhydrase proteins”. However, the authors also refer that, as it was already published, gamma carbonic anhydrases were found as part of complex I or suggested to be part of complex I in several other eukaryotes. Thus, I think that continue calling them as “plant-specific” is not correct. In fact, this topic was discussed several years ago and the consensus was to name them as “ancestral subunits”. I am not quite sure whether this name is the appropriate name because this encloses the idea that in Opisthokonta the domain was lost and we are not absolutely sure whether this domain was lost or never existed in this particular lineage. However, the name of “ancestral subunits” was accepted and certainly it is better than “plant-specific” which we gave at the beginning. Thus, in my opinion, the last sentence of the abstract and manuscript does not represent really a novelty because it was already proposed.

We thank the reviewer for this comment. We agree that carbonic anhydrase should be termed “ancestral subunit” and will change it accordingly in the text. It is however in plants that this ancestral organization is visualized to high resolution for the first time.

2-Lane 89 Carbonic anhydrase: specific component of the plant complex I. It is a sub-title; however it is not specific of plants. Thus, I suggest changing the sub title. The subtitle was changed accordingly.

3-The authors confirm the proposition of heterotrimeric composition of the CA domain, containing two γ CAs and one γ CAL, proposed in several papers; particularly it is shown that γ CA1 and two γ CAL subunits were found. This particular arrangement of two identical γ CAs and one γ CAL was proposed in Córdoba et al., 2019 (Plant Cell Physiol. 60:986-998). Moreover, the data of this publication suggest that the composition varies throughout development, being the CA2-CA2-CAL heterotrimer the more important in adult plants, mainly leaves and roots (Brassica and Arabidopsis contain exactly the same kind of γ CA subunits with high homology. Something about the conservation of the subunits between both species should be also appropriate as supplementary material, for example). The CA1-CA1-CAL heterotrimer is more important in embryos at the beginning of embryogenesis in Arabidopsis. Since mitochondria were isolated from *B. oleracea* inflorescence which are arrested flower meristematic tissue, did the author obtain data from other tissues? Did the authors find some heterogeneity at this level? Indeed, the identification of these subunits according to the text, figures and supplementary data appears to be very specific and clear. However, in the supplementary data, it is shown γ CA2, γ CA1 and γ CAL1. Why are there two γ CA1 subunits i

the model (Figure 2)? Are there models with γ CA2 subunits instead γ CA1 as well? In the extended data Figure 5 d and e, the authors show some indications why the protein detected is γ CA1 and not γ CA3 or γ CA2, however the authors wrote, "Still for the second γ CA (chain q) the scant density did not allow to clearly distinguish between γ CA1 and γ CA2, and was therefore built as γ CA1". The conclusion is not very clear to me although for the chain p was clear that γ CA1 was present. The writing should be changed, in my opinion to avoid confusion.

Brassica has also two γ CAL proteins. For the biochemical data obtained in Arabidopsis, apparently they are always at the same level and in all tissues, without preferences for heterotrimer formation. Then I would expect some heterogeneity although γ CAL1 and γ CAL2 are very similar as the authors mention in the legend of extended data of Figure 5c. Is it possible to guarantee the identity of this subunit (γ CAL1 or γ CAL2?) or are they so similar that they could be confused?

This is an interesting comment. Concerning the carbonic anhydrase composition, only flowers of *B. oleracea var botrytis* (cauliflower) were used during this study. That tissue was chosen for its easy access and the high yield of mitochondria purification, as well as cauliflower high sequence similarity to the model plant Arabidopsis. It would indeed be interesting to carry out structural analysis of complex I from different tissue/development stage, however the limiting step being mitochondria purification, we believe this would be an even more challenging project. Hence, we cannot address this question.

Concerning the subunit composition, γ CAL was clearly distinguished from γ CA, and one of the γ CA, could clearly be attributed to CA1 (chain p). However, there is some uncertainty regarding the second γ CA (chain q) (as pointed out in ED fig5 legend). This subunit was confirmed to not be γ CA3. However, it was not possible to clearly distinguish between γ CA1 and γ CA2. This could be due to compositional heterogeneity. This has now been also addressed in the main text.

As for the γ CAL subunit, to answer reviewers 2 and 3 comments, we now present the complete sequence alignment (excluding the target sequence) of γ CAL1 & 2 from both Arabidopsis and *B. oleracea var oleracea* to indeed highlight the very high sequence identity (98% pairwise identity and 99.8% BLSM62 pairwise positive) of these proteins, even across species. As pointed out originally in ED Fig 5 legend, given the high similarity of the two proteins and previous genetic screens both γ CAL1 and γ CAL2 could be there, and we chose to build γ CAL1 in the density.

4-Lane 115. The authors concluded that the CA domain of complex I is indeed active as a carbonic anhydrase because they found a density most likely corresponding to -HCO_3 bonded together with a Zinc atom to the active site. In my opinion this sentence should be modified. The binding to bicarbonate/ CO_2 was reported previously by Martin et al., 2009, (FEBS Lett. 583:3425-3430). However, the presence of bicarbonate does not mean necessarily that the enzyme is active as carbonic anhydrase but as a bicarbonate binding protein for regulation or to facilitate bicarbonate transport. This activity (i.e. Regulate access of cytosolic bicarbonate in the case of cyanobacteria) was suggested for CcmM protein of the shell of carboxysome in cyanobacteria which share high sequence and structural homology to gamma type carbonic anhydrases and binds bicarbonate but is inactive as a carbonic anhydrase (Cot et al., 2008 J Bacteriol. 190:936-945). Bicarbonate was suggested to enter the carboxysome in a regulated manner and then to be converted to CO_2 by a beta type carbonic anhydrase (CcaA). CcmM also binds RubisCO inside the carboxysome. The structure of the complex formed by CcmM and RubisCO was solved in Wang et al., 2019 (Nature 566:131-135). In 2010, it was discovered that some cyanobacteria species which do not contain a CcaA inside

the carboxysome, the CcmM protein was active as carbonic anhydrase (Peña et al., 2010, Proc Natl Acad Sci U S A. 107:2455-2460). Thus, I think that the conclusion is not supported by the data and therefore should be revised.

The results that we obtained here show that two out of three sites of the enzyme are inactive, they are now also shown in ED Fig. 6. However, from a structural viewpoint, the third site that we describe, at the interface of the two γ CA subunits, appears to be functional. Hence, the logical conclusion is that this one site could perform the activity. We toned down our conclusion in the main text, but we still believe that this one site could perform the activity.

5-Lane 119. The authors refer to an experiment in which γ ca2 mutants were complemented with inactive variants of γ CA2. They cite a review of Fromm et al. 2016 (Physiol Plant. 157:289-296). but I think this citation should be changed for the original paper of Fromm et al., 2016a (New Phytol. 211:194-207). However, in my opinion this experiment means that carbonic anhydrase activity is not essential but it cannot be concluded that binding to bicarbonate is not essential because in the mutant variants, aminoacids important for bicarbonate binding which are not so clear were not mutated. It would be interesting whether with the presented data the authors can determine what aminoacids are in contact with the bicarbonate molecule. This would add novel important information to the work.

Amino acids surrounding the bicarbonate were already shown in the Fig. 3 but are now described in the figure legend.

6-The discovery of the lipid complex interacting with the CA domain is very interesting and represents important data for future work. However, the authors attribute the reason why the CA mutants show strong phenotypes to the interaction with this particular lipid complex (I suppose the strong phenotype referred is lethal phenotype in normal growth conditions showing no complex I formation in ca1ca2 and ca1ca3 double mutants, Fromm et al., 2016a New Phytol. 211:194-207, Fromm et al., 2016b, J Exp Bot. 67:3079-3093, Córdoba et al., 2016, J Exp Bot. 67:1589-1603 and Córdoba et al., 2019, Plant Cell Physiol. 6:986-998). This is maybe true, but in my opinion is one of the reasons. The other more important reason according to the proposed assembly mechanism (Ligas et al., 2019 already cited) is that the CA domain is in the first detected intermediate of 200 kDa. Is it possible that in this first 200 kDa intermediate, this particular lipid complex is already attached and allows the binding of Nad2 subunit?

We don't have a clear answer to that. The 200kDa intermediate was not characterized in our study, thus one can only speculate if the lipids are already present. However, even though to lower resolution, we could also observe the lipid complex in the GLDH-bound assembly intermediate shown in our study, which hints at the importance of these lipids.

7-It is very interesting the confirmation of the plant-specific assembly intermediate containing the GLDH assembly factor which is not present in the full complex I.

We than the reviewer for emphasizing this finding, we were also quite happy to find this complex in our samples!

Minor points

Lane 31 "...composed cardiolipin and phosphatidylinositols." It should be "composed of"
Lane 50 The citation is not correct. The text corresponds to the Dudkina et al., which is cited immediately above (citation 11). The citation 12 is Sunderhaus et al., which is the first correlation that the identified gamma CAs formed the extra domain in plants.
Lane 98, the citation 18 is not corresponding with the text. For articles about Metanosarcina

thermophila Cam the authors could cite Alber et al., 1999, Biochemistry 38:13119-13128 and Tripp et al., 2004, J Biol Chem. 279:6683-6687. The citation of Parisi et al., 2004, should be cited when the authors mention about the presence of gamma CAs as complex I subunits. For citation 19, I suggest to add Perales et al., 2004, Plant Mol Biol. 56:947-957 in which the original data were published. In this paper heterotrimeric CA domain containing a CAL subunit was first proposed.

Lane 121-123. I think the authors should indicate that the carbonic anhydrase associated to photosynthetic complex I is in cyanobacteria which evolved more sophisticated carbon concentration mechanisms than plants.

Lane 123 instead "our complex I" please write mitochondrial complex I

Lane 166, the sentence of the conclusion is misreading.

Lane 172, change "but not in" by "except in"

All minor points have been addressed.

Eduardo Zabaleta. Instituto de Investigaciones Biológicas, IIB-CONICET-UNMdP, Mar del Plata, Argentina

Reviewer #3

The EM structure determination appears solid overall, although it is difficult to judge in the complete absence of any raw data and some typical quality measures that are typically provided. The model building into the 3.5-3.8 Å resolution maps was greatly helped by available models of the yeast and mammalian complex-I homologs. It is somewhat unfortunate that the model was built for Arabidopsis proteins in the absence of well-annotated cauliflower sequences. Overall, the most interesting insights of the paper are structural data about the interference of (plant-specific) GLDH with complete complex-I assembly by competition with (plant-specific) P1 binding, and the association of CA to complex-I. Weaknesses of the manuscript are the absence of any biochemical or genetic experiments as well as the model building based on another sequence. For publication, the authors should either model the cauliflower proteins (preferred) or substantiate the expected sequence variations for the modeled proteins compared to Arabidopsis.

We thank reviewer 3 for these comments.

Major points

- It appears odd that the authors modeled complex I subunits based on Arabidopsis sequences, while mass-spectrometry could detect the actual cauliflower proteins. The possibility to perform protein identification by MS strongly suggests that the protein database compiled for MS identification could be used for model building as well.

The database presented in supplementary data compile proteins from the Uniprot database corresponding to *B. oleracea* var. *oleracea* which is the wild cabbage and was falsely annotated "Cauliflower" in the excel file. As described in the Material & Methods section, to this date, there is no *B. oleracea* var. *botrytis* (cauliflower) protein sequences in the Uniprot (except for some few chloroplast-encoded proteins), and the closest fully sequenced member of the family (*Brassica oleracea* var. *oleracea*) is poorly annotated, this is why we chose to use Arabidopsis sequences. This strategy (purify from Cauliflower and built Arabidopsis proteins) was previously used to build the structure of the plant mitochondrial ribosome with great success. The misleading "cauliflower" mention in the excel file has been corrected, and material and method section has been extended on that matter.

- Cauliflower and Arabidopsis are reported to be 'closely related' – however, no approximation of expected sequence identities is provided.

- The assignment of different paralogs of CA in the EM density appears unreliable: the assignment bases on identification of characteristic side chains – however, the precise sequences of the cauliflower proteins are not even used. Neither local resolution of the EM map nor possible sequence variations suggest that paralogs can be distinguished.

- Extended data figure 3 a-d: without an indication of sequence conservation the in-depth visualization of side chains does not appear warranted despite the clear visibility of large side chains

Sequence conservation between Arabidopsis (being a Brassica) and other Brassica species (including cauliflower) is approximately of 90% sequence identity and has been added to the material & method section. This is now highlighted in ED Figure 3 with the CAL1 and CAL2 complete sequence alignment between Arabidopsis and *B. oleracea* var *oleracea* proteins, showing a 98% pairwise identity and a 99,5% BLOSUM62 pairwise positive. Sequence comparison between Arabidopsis and *B. oleracea* var *oleracea* is now shown for the CA subunits zones that are used for identification, showing that the chosen aminoacids for subunit distinction are invariable between these two species.

- Local resolution in a surface representation are standard for large assemblies determined by cryo-EM and not provided here

Local resolution of the final reconstructions are already provided in ED Fig2.

- Cross resolution of map and atomic map should be provided as a sanity test of atomic model and resolution

Map vs model FSCs, extracted from PHENIX validation, are now provided in ED Fig2.

- A representative micrograph must be placed in supplement

A representative micrograph has been placed in ED Fig1.

- Some idea of the unmasked complex (including the micelle) should be provided as supplement

The unmasked complex, where the micelle is displayed, has been added to ED Fig. 1.

- Some more detailed discussion of the analogies of animal vs plant subunits would be appreciated. E.g., how do P1 and NDUFB5 compare?

A comparison of P1 and NDUFB5/NUNM (which is absent in plants) is now provided in ED Fig. 5.

- I. 155: what does 'sufficient to attribute the density to the protein' mean. What is the criterion for 'sufficient'? what is near-atomic resolution (atomic would be better than ~2 Å?!).

For the attribution of GLDH to the density, as described in the text, even though the resolution was intermediate for most of this zone (~6 Å), the overall shape and size of the density corresponded to the generated homology model of GLDH. The observed position of GLDH is also in agreement with previous biochemical studies (Schimmeyer et al. 2016 Plant Mol. Biol. ; Ligas et al. 2019 Plant J.). Importantly, the N-terminal part of GLDH directly interacting with complex I presents a higher resolution sufficient to distinguish the bulky side-chains of this domain (shown in Fig. 3), thus confirming our initial conclusions. Near-atomic resolution was removed from the text.

- I. 157: what does 'slightly displaced' mean?

Both for the CA domain and B14.5b a more precise (numbered) description of the movements have been added to Fig. 3 and figure legend.

- L 166/167: sentence is incomplete – and unclear. What are 'most likely ancestral features of this respiratory complex'?

The sentence has been corrected

- Short introductory sentences on P_D and P_P module would be appreciated by a broad audience.

Short introductory sentences have now been included in the text. "The membrane part, composed of the proximal (P_P) and distal (P_D) modules relative to the matrix arm, where proton pumping takes place,..."

- Universal role of carbonic anhydrase is somewhat vaguely suggested in the abstract, but not really explained. Also the term 'ancestral-like' organization is unclear.

As pointed out by reviewer 2, several bioinformatic and biochemistry approaches showed that carbonic anhydrase is part of the complex I in possibly all eukaryotes except opisthokonta (due to the loss of the enzyme in this group). Thus, the organization of the plant complex I presented here (with the carbonic anhydrase domain) would represent an "ancestral" form of the complex, that was not observed previously to this resolution, as only complex I from opisthokonts (yeast and animals) have been characterized to date. 'Ancestral-like' was replaced to ancestral according to reviewer 2 comments, as this is an accepted term in the field, we hope reviewer 3 concurs as well. The "role" of the carbonic anhydrase is actually not well understood, the activity of the enzyme is not required, however its presence is crucial for complex I early assembly steps, as indicated by several studies already (Refs).

minor points

- L. 158: 'and 15 kDa'? something appears to be missing.

- P_D module should be P lowercase D consistently.

- L. 68: it should probably read: "...the NADPH molecule in the 39kDa subunit, ".

All minor points have been addressed

Altogether, we wish to thank all the reviewers for their constructive comments that lead to improving our manuscript presentation and content.

We hope that our revised version of the manuscript will be acceptable for publication by Nature Communication.

With best regards,

Yaser Hashem and Florent Waltz

REVIEWER COMMENTS

Reviewer #1 (Remarks to the Author):

In the revised manuscript, Soufari et al have addressed major points raised by the reviewers but in my opinion have not added more substance to the manuscript. Though it was not explicitly mentioned in the previous referees report (it was hinted that the manuscript had few salient features by the reviewers), they could have expanded a bit more on the general description by comparing the plant complex against other complex I structures, whether the current class they have fits with active/deactive states (or closer to yeast complex), hypothesis on how GLDH might be dislodged to allow the binding of P1 – these can be added as discussion. In particular the later will be of great interest. As it reads, the main text is very thin and would benefit from some more discussion on salient features. This is important because most readers will not look at the table and go through the subunits and cross check with other homologues. At the moment, only researchers working on plant mitochondrial complex will be able to understand and not general readers.

Other points:

In data statistics table – magnification is 36,000 or simply 36000 and not 36.000

Final particle images for assembly intermediate is missing

There are 3 numbers for B factors, which I am assuming is low/high and average. If this is correct, please add a footnote.

In ED Figure 1 – scale bar to be added for the micrograph and please mention the box size used for class averages

The description of the structure building and model refinement needs a careful look and editing (like 396-397). It is still not clear if the reciprocal space refinement was done like the protocol used for X-ray data. Also, the B-factor values for some of the model seems very low (4.9/0.35) – not sure if this is because of mask used during refinement.

Another recommendation, is to do cross-validation of the model/map refinement (Phenix probably has this routine and if it was done already then it should be mentioned). Even otherwise it is relatively straight forward to do (Brown et al 2014 Acta D).

The density for horizontal helix of Nad5 (figure 1) seems disjointed, is this because of contour level (the complete model is built – panel b). Please comment.

In methods, line 364, the pixel size should be 1.11 Å?

In methods, line 405, the heading says mitochondrial ribosome and not Complex I!

Reviewer #2 (Remarks to the Author):

The revised manuscript entitled "Specific features and assembly of the plant mitochondrial complex I revealed by cryo-EM" by Soufari et al. was improved respect to the original version and addresses all my concerns. I want only to do one last comment. The authors added the following new sentence concerning a possible role if the CA proteins were not active as carbonic anhydrases. "even if a role in HCO₃ fixation cannot be excluded, similarly to what is observed in cyanobacterial CcmM22". I much appreciate they taken into account my observation, however, in my opinion, the world fixation is not the more appropriate. I would write this sentence as follow: "even if a role in HCO₃ transport cannot be excluded, similarly to what is proposed for the homologue

cyanobacterial CcmM22".

This work represents an important contribution for mitochondrial biology field.

Reviewer #3 (Remarks to the Author):

The authors addressed all points raised by this reviewer adequately. The work is of high interest for mitochondrial evolution and high technical standard. Hence, the work should be published.

Response to reviewers' comments:

Reviewer #1 (Remarks to the Author):

In the revised manuscript, Soufari et al have addressed major points raised by the reviewers but in my opinion have not added more substance to the manuscript. Though it was not explicitly mentioned in the previous referees report (it was hinted that the manuscript had few salient features by the reviewers), they could have expanded a bit more on the general description by comparing the plant complex against other complex I structures, whether the current class they have fits with active/deactive states (or closer to yeast complex), hypothesis on how GLDH might be dislodged to allow the binding of P1 – these can be added as discussion. In particular the later will be of great interest. As it reads, the main text is very thin and would benefit from some more discussion on salient features. This is important because most readers will not look at the table and go through the subunits and cross check with other homologues. At the moment, only researchers working on plant mitochondrial complex will be able to understand and not general readers.

We thank reviewer 1 for his/her positive appreciation of our work. Based on your advices, discussions have been extended in the main text.

Other points:

In data statistics table – magnification is 36,000 or simply 36000 and not 36.000

Has been modified accordingly.

Final particle images for assembly intermediate is missing

This is now indicated.

There are 3 numbers for B factors, which I am assuming is low/high and average. If this is correct, please add a footnote.

This is now indicated.

In ED Figure 1 – scale bar to be added for the micrograph and please mention the box size used for class averages.

A scale has been added accordingly, and the box size (1.11*360pix) for class average is indicated in the methods section.

The description of the structure building and model refinement needs a careful look and editing (like 396-397). It is still not clear if the reciprocal space refinement was done like the protocol used for X-ray data. Also, the B-factor values for some of the model seems very low (4.9/0.35) – not sure if this is because of mask used during refinement.

The method section for structure building has been corrected. B-factor values have been checked and corrected if necessary.

Another recommendation, is to do cross-validation of the model/map refinement (Phenix probably has this routine and if it was done already then it should be mentioned). Even otherwise it is relatively straight forward to do (Brown et al 2014 Acta D).

CC numbers (phenix output) have been added to ED figure 2.

The density for horizontal helix of Nad5 (figure 1) seems disjointed, is this because of contour level (the complete model is built – panel b). Please comment.

This is indeed due to the contour level. Nad5 C-terminal transverse helix (in the Pp module) is visible, as well as the rest of the protein constituting the majority of the Pd module. Resolution of the horizontal helix, linking the final transverse helix and the Pd module part of the protein is scatter, thus it appears disjointed in the map. This is something that has been observed previously for the human and yeast complexes.

In methods, line 364, the pixel size should be 1.11 Å?

This is now corrected.

In methods, line 405, the heading says mitochondrial ribosome and not Complex I!

This is now corrected.

All points have been addressed.

Reviewer #2 (Remarks to the Author):

The revised manuscript entitled “Specific features and assembly of the plant mitochondrial complex I revealed by cryo-EM” by Soufari et al. was improved respect to the original version and addresses all my concerns. I want only to do one last comment. The authors added the following new sentence concerning a possible role if the CA proteins were not active as carbonic anhydrases. “even if a role in HCO₃ fixation cannot be excluded, similarly to what is observed in cyanobacterial CcmM22”. I much appreciate they taken into account my observation, however, in my opinion, the world fixation is not the more appropriate. I would write this sentence as follow: “even if a role in HCO₃ transport cannot be excluded, similarly to what is proposed for the homologue cyanobacterial CcmM22”. This work represents an important contribution for mitochondrial biology field.

Again, we thank reviewer 2 for his/her positive appreciation of our work. The sentence has been modified accordingly.

Reviewer #3 (Remarks to the Author):

The authors addressed all points raised by this reviewer adequately. The work is of high interest for mitochondrial evolution and high technical standard. Hence, the work should be published.

We thank reviewer 3 for his/her positive appreciation of our work.